# Challenges of BTV-Group Specific Serology Testing: No One Test Fits All

**DOI:** 10.3390/v16121810

**Published:** 2024-11-21

**Authors:** Antonio Di Rubbo, Kalpana Agnihotri, Timothy R. Bowden, Michelle Giles, Kimberly Newberry, Grantley R. Peck, Brian J. Shiell, Marzieh Zamanipereshkaft, John R. White

**Affiliations:** The Commonwealth Scientific and Industrial Research Organisation (CSIRO), Australian Animal Health Laboratory, Australian Centre for Disease Preparedness, 5 Portarlington Road, East Geelong, VIC 3219, Australia; kalpana.agnihotri@csiro.au (K.A.); timothy.bowden@csiro.au (T.R.B.); michelle.giles@csiro.au (M.G.); kim.newberry@csiro.au (K.N.); grant.peck@csiro.au (G.R.P.); jbshiell@gmail.com (B.J.S.); marzieh.zamanipereshkaft@csiro.au (M.Z.); john.white@csiro.au (J.R.W.)

**Keywords:** BTV, BTV-15, sero-surveillance, validation, competition ELISA, sandwich-format ELISA

## Abstract

A newly formatted enzyme-linked immunosorbent assay (ELISA) for the detection of antibodies to bluetongue virus (BTV) was developed and validated for bovine and ovine sera and plasma. Validation of the new sandwich ELISA (sELISA) was achieved with 949 negative bovine and ovine sera from BTV endemic and non-endemic areas of Australia and 752 BTV positive (field and experimental) sera verified by VNT and/or PCR. The test diagnostic sensitivity (DSe) and diagnostic specificity (DSp) were 99.70% and 99.20%, respectively, for bovine sera, and 97.80% and 99.50%, respectively, for ovine sera. Comparable diagnostic performances were noted for the sELISA compared to four competition ELISAs. While the sensitivity of the sELISA remained unaffected by BTV-15 positive sera, the cELISAs were not as sensitive. BTV-15 is endemic to Australia, and early warning depends on sensitive diagnoses of all serotypes: endemic or incurring. The sELISA failed to discriminate against epizootic hemorrhagic disease virus (EHDV) antibodies, the most serologically related orbivirus to BTV. The ACDP cELISA and the IDEXX kit showed cross-reactivity with some EHDV serotypes, with the least cross-reactive being the VMRD and the IDVet kits. Cross-reactivities, however, were also detected in sera raised experimentally from 10 isolates of the 21 known non-BTV orbiviruses. In this case, the sELISA was the least affected, followed equally by the VMRD and IDVet kits, and the IDEXX kit and the ACDP cELISA were the least discriminatory. In addition to exclusivity assessment of the ELISAs, an inclusivity assessment was made for all ELISAs using well characterized reference sera positive for antibodies to all serotypes BTV-1 to BTV-24.

## 1. Introduction

Bluetongue virus (BTV) is the prototype member of the genus *Orbivirus* within the family *Sedoreoviridae* [1]. BTV is one of 22 recognized species in the genus. Currently, there are 27 recognized BTV serotypes [2,3,4]. Bluetongue (BT) is primarily a vector-borne viral disease that affects wild and domestic ruminants and causes a haemorrhagic disease, mainly targeting the endothelial lining of blood vessels, resulting in a variety of signs that often leads to a moribund state and sometimes death [4].

BTV is endemic in many tropical and sub-tropical countries and is typically transmitted to susceptible ruminants via blood feeding by *Culicoides* spp. midges The increasing global range of BTV infection is now evident, as traditional northern and southern most limits of BTV distribution have expanded with incursions into Europe, Australia, America, Asia and other countries that had never previously reported BTV infections [4]. In susceptible ruminants of economic importance, BT disease can cause reduced fertility rates and milk production, decreased wool quality and yield, and mortality. Prolonged loss of productive animals due to BT outbreaks causes economic hardship from loss of animal product sales and detection alone of BTV in export livestock can have severe economic impacts on countries reliant on live animal and animal product exports [5]. As a result, BTV has been listed by the World Organization for Animal Health (WOAH) as a notifiable disease and outbreaks in previously BTV undetected zones result in movement and trade restrictions for those regions.

Infection in cattle is typically subclinical, although higher incidence and severity of clinical disease can occur in naïve cattle, for example with BTV-8 [6] and BTV-3 [7]. As sheep display varying levels of susceptibility to BT disease, BTV infections of livestock can occur unobserved and only be detected by active surveillance [8].

WOAH-approved methods (2021; Chapter 3.1.3) for confirmation of clinical infection with BTV in animals include pan-RT-PCR and RT-qPCR. PCR is also useful for assessing freedom from infection prior to movement [3]. Approved methods for sero-surveillance and population freedom from infection testing prior to movement include various cELISAs platforms. Competition ELISAs are not recommended for confirmation of clinical cases as they only detect antibody from day 7 to 14 post infection. Notably, there is significant immunological cross reactivity among members of the BTV serogroup when using serological methods, such as ELISA. Virus neutralization tests (VNT) are used to differentiate between serotypes, but cross reactivity can still occur with multiple, even single, BTV infections. Despite these limitations for specific serotype identification and acute phase infection detection, systematic routine serological screening for BTV-specific antibodies remains an essential component of a fully comprehensive approach to early detection of novel BTV incursions. Countries now routinely employ groups of ‘sentinel’ animals, strategically placed near the interface of known endemic and potential epidemic regions, that are regularly screened for virus specific immune responses [9,10].

Competition/blocking ELISA formats employing monoclonal antibodies (mAbs) specific for epitopes on conserved viral structural proteins have become the preferred alternative for BTV-specific antibody detection [8] and are the prescribed test for international trade by WOAH. The BTV outer capsid VP2 protein and inner core VP7 proteins are highly immunogenic and are targets of most serological assays. While VP2 provides information on BTV serotype, and is responsible for elicitation of neutralising antibody responses, the major core protein VP7 is most often targeted for serogroup identification of BTV infection, despite having a relatively low degree of sequence conservation compared to other capsid proteins [11]. Caution must therefore be exercised where monoclonal antibodies form the basis of detection, as variations in VP7 sequences are known to exist and may affect assay sensitivity [12].

Competition ELISAs indirectly detect virus-specific antibodies by competing with them for a single (virus-specific) epitope using a monoclonal antibody (mAb). This increases test specificity and results in fewer false positive reactions, but also reduces BTV serogroup sensitivity where antibodies to specific serotypes lack or have reduced affinity to the target epitope [13,14]. Many existing ELISAs, whether in-house or commercially developed, have proven inadequate for the detection of some BTV serotypes, particularly BTV-15, which has been detected in Australia, Africa, China, Cyprus, Israel, and South Korea [15,16,17,18,19]. This may result from evident genomic differences between BTV-15 and other BTV strains [20,21].

Indirect ELISAs, for direct detection of virus-specific antibodies, have been less enthusiastically adopted in recent years due to cross reactivity with related viruses, reducing the specificity of these tests. This paper describes a newly developed and validated test, the BTV sandwich-format ELISA (sELISA), that uses a rabbit polyclonal anti-BTV antiserum as an antigen-capture system to trap recombinant BTV VP7 protein, which in turn traps anti-BTV VP7 antibodies present in test sera. The final step detects host multivalent anti-BTV VP7 polyclonal antibodies using a species-specific antibody conjugate (Figure 1).

Australia’s complex ecosystem includes several orbivirus species that share features with BTV and antibodies to these viruses have been detected in cattle and sheep. These include EHDV, Corriparta virus [22], Paroo River virus [23], Wallal virus, Warrego virus [24], Middle Point virus [25], Palyam virus, Bunyip Creek virus, Marrakai virus [26], Eubenangee virus [27,28] and Tilligerry virus [29,30]. Furthermore, Jeggo et al. (1983) [31] and Della-Porta et al. (1985) [32] showed that occurrences of multiple infections with a variety of orbiviruses in the field may result in the development of heterologous immune responses to orbiviruses not encountered by the host animal. There is, therefore, a clear need to investigate potential cross-reactivity to antibodies directed against other orbiviruses. When compared with an ACDP cELISA [14] and three commercially available BTV ELISA kits the sELISA showed increased sensitivity to all BTV serotypes assessed. The sELISA also showed increased specificity by reduced detection of several circulating orbiviruses and arboviruses in Australia. This assay failed, however, to discriminate against EHDV antibodies, whereas the other ELISAs tested demonstrated varying degrees of ability to avoid EHDV specific antibody detection.

Validation data for the newly developed sELISA, in comparison with an ACDP cELISA [14], and three commercially available BTV ELISA kits, are presented, with each ELISA being assessed for analytical performance characteristics, including analytical sensitivity and specificity, and diagnostic sensitivity and specificity. Specificity assessments included a robust selection of sera to assess each ELISA’s selectivity, inclusivity and exclusivity.

## 2. Materials and Methods

### 2.1. Sera

Experimental antisera produced in cattle and sheep to Australian BTV serotypes and to other arboviruses were prepared as described previously [20,33,34] (Table 1 and Appendix A). Reference sera were obtained from the Onderstepoort Laboratory, South Africa, and ACDP. Inactivated ovine sera to South African BTV serotypes were obtained from the Commonwealth Serum Laboratory (CSL), Parkville, Australia (Table 1). Field SNT- and/or PCR-positive sera were from diagnostic sample submissions collected from cattle and sheep as part of the National Arbovirus Monitoring Program (NAMP) and the Northern Australia Quarantine Strategy (NAQS). BTV negative bovine and ovine sera from Victoria were supplied by Agriculture Victoria or collected from non-endemic areas of Australia as diagnostic sample submissions. Pre-bleed sera for experimental work collected from Victorian animals and virus neutralization test (VNT)-confirmed negative sera from inside the BTV endemic zones were also included. The BTV-15 positive sera consisted of 12 field samples from 2013, 1 field sample from 2019, and 1 Australian and 1 South African reference experimental serum. The 14 PCR positive bovine field samples (2014–2024) were obtained from BTV diagnostic surveys.

### 2.2. ROC Analyses

The receiver operating characteristics (ROC) curve analyses and cut-off determinations were performed using MedCalc Statistical Software version 23.0.6 (MedCalc Software Ltd., Ostend, Belgium; https://www.medcalc.org; 2024).

VNT and/or PCR sera were confirmed positive for several BTV serotypes. For the validation of the ACDP BTV cELISA, positive samples included 315 bovine post-infection sera, 81 ovine samples that included 30 post-infection sera, and 51 experimentally infected ovine sera. The negative samples consisted of 650 bovine and 386 ovine sera that were sourced from BTV-free zones or from endemic areas, the latter confirmed to be BTV-negative by VNT.

For validation of the BTV sELISA, positive samples included 661 naturally infected bovine and 91 ovine sera, of which 51 were experimentally infected and confirmed by VNT. Negative samples included 509 bovine and 440 ovine sera or plasmas.

### 2.3. BTV VP7 Antigen Production

BTV VP7 antigen used in both the sELISA and the cELISA was prepared using the Bac-to-Bac system (ThermoFisher Scientific™, Scoresby, Australia) as described by the manufacturer. In brief, pFastBac1 containing the coding sequence for BTV1 (CSIRO 156) VP7 was used to transform DH10Bac *E. coli*. Bacmid was purified using a PureLink HiPure plasmid kid (Thermo Fisher Scientific™, Scoresby, Australia), combined with Cellfectin II in Grace’s medium and used to transfect Sf21 cells (1 µg/8 × 10^5^ cells). Passage 1 baculovirus was collected and used to amplify virus stocks through 2 sequential infections using a multiplicity of infection (m.o.i.) of 0.1 with Sf21 cells in suspension. BTV1 VP7 protein was produced by infecting 1 L Sf21 cells at ~2 × 10^6^ cells/mL with recombinant baculovirus at 5 × 10^7^ pfu/mL (assumed) giving an m.o.i. of 1. Infected cells were cultured for 3 days at 28 °C with orbital agitation at 100 rpm then harvested by centrifugation at 600× *g* for 10 min at 4 °C. Cell pellets were washed twice with PBS, resuspended in 25 mL PBS containing protease inhibitors (Sigma-Aldrich^®^, Bayswater, Australia) and lysed by sonication. Lysate was clarified by centrifugation at 10,000× *g* for 10 min at 4 °C and supernatant containing BTV1 VP7 used as the final antigen preparation.

### 2.4. BTV Sandwich ELISA (sELISA)

Nunc Maxisorp ELISA immuno-plates (Thermo Fisher™, Rofkilde, Denmark) were coated with 50 μL/well of polyclonal rabbit anti-BTV 20 sera [35] at 1/4000 in 0.05 M hydroxymethylaminomethane (Tris) pH 9.0 (Sigma-Aldrich^®^, Truganina, Australia) coating buffer. Plates were incubated at 37 °C for 1 h with shaking. Plates were treated with 200 μL of blocking buffer: 2% (*w*/*v*) Diploma^®^ skim milk powder (SMP) in ddH_2_O. Plates were incubated at 37 °C for 10 min with shaking, then washed 3 times in phosphate buffered saline (PBSA), pH 7.2 with 0.05% (*v*/*v*) Tween 20 (Merck, Bayswater, Australia) using an automatic plate washer (Skatron Instruments, Skan-Washer 300, Thermo Fisher™, Singapore). Fifty microliters of BTV VP7 antigen diluted 1/2000 (as determined by checkboard titration) in ELISA sample diluent (SD: PBSA + 0.05% (*v*/*v*) Tween 20 + 1% (*w*/*v*) SMP) were added to all wells except to antigen-free mock control wells, which received 50 µL of ELISA diluent only. Plates were incubated at 37 °C for 30 min with shaking. During this time, test sera, BTV negative bovine or ovine serum (diluted 1/100 in SD) and BTV positive bovine or ovine serum (titrated from 1/100 to 1/6400 in SD) were prepared. Plates were washed as previously described and 50 µL of each test sample or control sera were added to all wells except blank control wells. Plates were incubated at 37 °C for 30 min with shaking, washed as previously described, and 50 µL of horseradish peroxidase (HRP)-conjugated goat anti-bovine IgG H+L (Invitrogen REF A18751 Frederick, USA) or donkey anti-sheep fragment antigen binding (Fab) (Novex^®^, Frederick, MD, USA) diluted 1/2000 in SD were added. Plates were incubated at 37 °C for 30 min with shaking followed by washing and addition to each well of 50 μL of 3,3′5,5′-tetramethylbenzidine (TMB) substrate (Sigma-Aldrich^®^, Melbourne, Australia^®^). Plates were incubated at room temperature and the reaction was stopped within 8 min with 1 M sulfuric acid (Ajax Finechem Wollongong, Sydney, Australia). Plates were read at 450 nm with a Multiscan plate reader (Thermo Fisher™, Singapore). Results were transformed using sample to positive control ratio (S/P). S/P ratio was calculated using the formula:S/P = (S − M)/(P − M)
where: S = average absorbance value of test sera wells; M = absorbance value of the mock (no antigen) well; P = average absorbance value of the positive control serum wells diluted at 1/1600.

### 2.5. ACDP BTV Competition ELISA (ACDP cELISA)

BTV cELISA was performed as previously described [10] with the following modifications. Nunc Maxisorp ELISA immuno-plates replaced polyvinylchloride (PVC) plates. Casein 10 × blocking buffer (Sigma Aldrich^®^, Saint Luis, MO, USA) diluted 1/10 in ddH_2_O (CBB) replaced 1% (*w*/*v*) SMP in PBST as the sample diluent. VP7 antigen replaced the P200 crude preparation of BTV-1. In addition, the preparation of BTV-15 antigen was not incorporated in combination with BTV-1 in the coating antigen mixture. The VP7-specific mAb, 20E9.B7.G2 used by Lunt et al. (1988) [14], was biotin-labelled and was followed by HRP conjugated streptavidin (Jackson Immuno-research Baltimore Pike, West Grove, PA, USA) diluted 1/16,000 in CBB.

### 2.6. Commercial ELISAs

The commercial assays used in this study were the IDEXX Bluetongue Competition Ab Test (IDEXX Mount Waverley, Melbourne, Australia), Bluetongue Virus Antibody Test Kit, cELISA v2 (VMRD, Pullman, WA, USA), and ID Screen Bluetongue Competition ELISA (IDVet, Innovative Diagnostics Grabels, Montpellier, France). All ELISA kits were used as described in the manufacturers’ instructions.

### 2.7. Virus Neutralization Test

Virus neutralization tests (VNT) were performed according to the method described in the WOAH Terrestrial Manual, 2021 (Chapter 3.1.3) [3]. In brief, sera were titrated four-fold from 1/10 to 1/640 in Earle’s salt Minimum Essential Medium (EMEM; Thermo Fisher, Brisbane, Australia) supplemented with 10 mM HEPES (pH 7.5); 2 mM glutamine; 0.8 mg/mL penicillin-streptomycin and 1 µg/mL amphotericin B (Fungizone^®^) (Sigma Aldrich^®^, Saint Luis, MO, USA). Titrations of sera were performed in 50 µL EMEM in duplicate wells of Nunc 96-well tissue culture plates (Thermo Fisher, Rofkilde, Denmark). Fifty microliters of 50% tissue culture infectious dose (TCID_50_) of each BTV serotype was separately added to all samples and dilutions. Plates were incubated at 37 °C for 1 h with 5% CO_2_ then 100 µL of African Green Monkey kidney epithelial cells (VERO) at 1.5 × 10^5^ cells/mL in EMEM, further supplemented with 10% fetal bovine serum, were added to all wells. Plates were incubated for 7 d at 37 °C. Plates were then fixed with 200 µL fixative solution (4% (*v*/*v*) formalin (UNIVAR analytical reagent, Wantirna South, Australia), 100 mM NaCl and 0.1% (*w*/*v*) methylene blue (Sigma Aldrich, Steinheim, Germany). Plates were fixed at RT for 24 h then washed under tap water. Titers for each serum were established using the Spearman-Karber method.

## 3. Results

### 3.1. Diagnostic Sensitivity (DSe) and Diagnostic Specificity (DSp): Setting of Thresholds—ROC Analyses

Median ACDP cELISA percent inhibition (PI) values for the negative population were less than 9%, and those of the positive population were 95% or above (Table 2). These were associated with a sufficiently narrow data spread to set thresholds with potentially acceptable resolution. PIs from bovine and ovine sera were used separately to optimize the threshold selection using ROC curves plotting sensitivity against (100—specificity) at incrementally altered positive/negative thresholds.

Median S/P values for the negative population were below 0.1 and those of the positive population were 1.6 for ovine and 7.3 for bovine or above (Table 2) and associated with a sufficiently narrow data spread to set thresholds with potentially acceptable resolution. DSe and DSp were optimized by setting the positive S/P threshold for bovine sera at >0.62 and for ovine sera at >0.36. These settings resulted in DSe of 99.7% (95% CI: 98.9–100.0%) and DSp of 99. 21% (95% CI: 98.0–99.8%) for bovine sera (Figure 2A) and DSe of 97.8% (95% CI: 92.3–99.7%) and DSp of 99.5% (95% CI: 98.4–99.9%) for ovine sera (Figure 2B).

The PI cut-off value for positive bovine sera set to ≥40% resulted in DSe and DSp of 96.2% (95% CI: 93.4–98.0%) and 94.0% (95% CI: 91.9–95.7%), respectively (Figure 2C). For the ovine sample population, the PI cut-off value for a positive serum was estimated at >45% with DSe and DSp of 95.1% (95% CI: 87.8–98.6%) and 97.1% (95% CI: 94.9–98.6%), respectively (Figure 2D). The two populations (bovine and ovine) were also analyzed in combination where a PI cut-off estimate for a positive sample of >41% resulted in DSe of 95.9% (95% CI: 93.5–97.7%) and DSp of 95.2% (95% CI: 93.7–96.4%), respectively (Figure 2E).

The spread of data for positive and negative populations for the ACDP cELISA and sELISA for both species shows the extent to which false negative and false positive values occur relative to selected thresholds (Figure 2). A comparison of two negative populations of cattle sera collected inside (P1) or outside (P2) the BTV circulation zones in Australia was performed by analyzing results for 104 P1 and 546 P2 sera using the cELISA and 81 P1 and 429 P2 in the sELISA. Differences in the PI or S/P ratio between populations were evident. Using the cELISA 34 of 104 P1 sera produced a PI value above the threshold of 40% (DSp = 67%), representing a 33% false positive rate. This compared to 5 of 546 P2 sera exceeding the threshold (DSp = 99.08%), representing a less than 1% false positive rate (Figure 3A). Using the sELISA 4 out of 81 P1 samples produced an S/P value above the set threshold of 0.62 (DSp = 95.06), representing a 4.94% false positive rate. No false positives were detected in the P2 population (Figure 3B). A *t*-test: Two-Sample Assuming Unequal Variances estimates analysis showed P1 ≠ P2 (Figure 3A,B).

### 3.2. Analytical Sensitivity

The limit of detection for the sELISA was compared against all ELISAs by titrating samples at starting dilutions appropriate to each assay (i.e., 1/100 for the sELISA, 1/10 for the ACDP cELISA, 1/5 for the IDEXX kit, and 1/2 for the IDVet and VMRD kits) followed by four-fold dilution to endpoint (the dilution at which the samples produced a negative result in that test). Three Australian field-positive bovine sera and two experimentally infected positive ovine sera were used in their respective bovine and ovine assays. Bovine sera assessed using the sELISA exhibited higher sensitivity than any other test (Table 3). For ovine sera, the sELISA proved to be more sensitive than the VRMD and IDVet tests, equally sensitive to the ACDP cELISA, and less sensitive than the IDEXX test (Table 3).

### 3.3. Comparing ELISA DSe and DSp Characteristics

Test performance analyses were conducted empirically for the ACDP cELISA and sELISA. Performance characteristics of the three commercial BTV cELISAs have been published previously [36]. The sELISA demonstrated improved or comparable DSe and DSp when compared with the other assays. Besides the sELISA, the most sensitive was the IDVet, followed by the ACDP cELISA, IDEXX, and VMRD.

All ELISAs produced, or were reported to exhibit ≥99% DSp except the ACDP cELISA, which was found to be >95.0% (Table 4). The ACDP cELISA exhibited similar diagnostic characteristics for ovine sera and bovine sera. Using cut-off values of ≥40% and ≥45% for bovine and ovine sera, respectively, the DSe and DSp were 96.2% and 94.0% for bovine and 95.1% and 97.1% for ovine sera. The assays showing the lowest levels of DSe were the VMRD test (69.50%), followed by the IDEXX test (82.8%) (Table 4).

Differences between the five tests in their ability to detect antibodies to different BTV serotypes were identified. Only the BTV sELISA detected all serotypes that were assessed.

No other assay adequately detected antibodies to BTV-15, but all other serotypes in the Australian panel were detected (Table 1). The same was true when using South African reference sera with the following exceptions: the IDVet test did not detect antibodies to BTV-14 and 19; the ACDP cELISA showed a weak reaction to BTV-15 but detected all other serotypes; the VMRD test missed BTV-5, 6 and 10; and the IDEXX kit failed to adequately detect BTV-7 (Table 1).

### 3.4. Sensitivity to BTV-15

Lunt et al. (1988) and (2009) [9,10], and this study, report that cELISA platforms available for BTV group testing demonstrate poor sensitivity for detection of antibodies to several strains of BTV, including BTV-15. Thirteen BTV-15-reactive bovine field sera, including two ovine reference BTV-15-positive antisera, were assessed using each of the ELISAs. Only the sELISA identified all 15 samples as being positive for the presence of BTV antibody (Table 5). The ACDP cELISA detected 9 positive samples (60%), followed by the IDVet assay which detected 3 positive samples (20%), and the IDEXX and VMRD assays each detected 2 positive samples (13% and 14%, respectively)—noting that one sample used in the VMRD assay was not able to be thoroughly scrutinized due to insufficient sample volume. A further 2 samples were deemed indeterminate using the IDEXX assay.

### 3.5. Sensitivity to Early Infections

Detection of nucleic acid can indicate current or recent exposure to BTV for up to four months post-infection (p.i.) in sheep [37] and up to five months p.i. in cattle [38]. Fourteen PCR-positive Australian cattle field samples were used to assess the ability of each ELISA to detect early BTV infections (Table 6). The sELISA detected BTV antibody in all the PCR-positive samples. The IDVet ELISA identified 13 of the samples as positive for anti-BTV antibodies, the ACDP cELISA and VNT detected antibody in 11 samples, the IDEXX assay identified 8 positives, and the VMRD proved to be the least sensitive detecting only 5 positive samples. No correlation between samples identified as negative and assay used was evident (Table 6).

## 4. Discussion

Competition ELISAs depend upon test serum antibodies inhibiting binding of a mAb to its unique epitope. This has the innate benefit of providing high test specificity, but this may come at the expense of test sensitivity. Changes in the targeted epitope of a circulating strain may result in a reduced or abrogated capacity for a cELISA to detect host antibodies raised against that strain. This is true for tests targeting orbiviruses where variants have arisen [39,40,41,42], including the BTV serogroup, where some serotypes can exhibit substantial strain-related antigenic variations. Consistent with this, BTV-15 antibodies were poorly detected by the cELISAs tested and all tests failed to detect various other BTV serotype antibodies, highlighting differences in the diagnostic characteristics of BTV serogroup assays. Failure to adequately detect antibodies to BTV-15 (and other strains) by widely available cELISAs presents a biosecurity risk that remains to be mitigated. Whilst not a prudent surveillance strategy, simultaneous circulation of multiple serotypes in overlapping geographic zones has fortuitously diminished this risk. For example, in the BTV-15 endemic region of the eastern Australian coast, co-circulation of BTV-15 with BTV serotypes 1, 16, and 21 has, but not completely, avoided false negative occurrences in recent years.

Changes to climate and weather patterns impact on novel geographic incursion and related epidemics of arbovirus dependent diseases across the globe. Resultant unprecedented increases in insect population numbers and their more extensive and prolonged dispersion into more northerly and southerly latitudes has resulted in recent regular waves of outbreaks in historically recognised ‘virus-free’ zones [43]. From approximately the mid-late 1990s onwards, Europe has experienced several bluetongue disease outbreaks of ever more extensive geographic incursion (involving a variety of different serotypes) into previously recognised BTV-free regions. Regular serological screening for increased BTV-specific antibody activity in sentinel animals can provide an early warning of new insect population incursion and blood feeding activity and quickly inform the need for targeted molecular diagnostic testing within a given region.

Inadequate detection of BTV-15 or threats represented by the potential introduction of novel strains with sero-phenotypic characteristics that may prevent detection using existing cELISAs was the basis for development of the sELISA described herein. The diminished serological sensitivity to BTV-15 exhibited by the cELISAs tested must lie in the capacity of the infected host to elicit an antibody response to a specific VP7 epitope that interacts in some way with the binding region of the VP7-specific mAb. The diminished serotype specific sensitivity may not be restricted to BTV-15.

In contrast to the cELISAs that rely on detection of antibody to a single VP7 epitope, the sELISA detects the full cohort of host antibodies that bind epitopes across the entire VP7 protein. This may improve sensitivity and explain why only the sELISA was able to identify all BTV-15 positive sera. Wang et al. (1994) [44] showed that the BTV-15 VP7 sequence diverged significantly from that of other members of the BTV serogroup, where amino acid sequences of BTV-15 VP7 showed sequence identities of 80 to 84% compared with 93 to 100% observed within all other serotypes. While it is beyond the scope of this study to determine the mechanisms underlying the reduced power of cELISAs to detect all BTV serotypes, epitope mapping of the mAbs used in the cELISA platforms would be informative when combined with information on the degree to which the identified epitopes are conserved across serotypes.

In this study, sera from animals in Australia were used for assessment of the sELISA and ACDP cELISA, and sera from animals in Europe were used for validation of the commercial cELISAs (as reported by Niedbalski et al., 2011 [36]). Positive sera for the commercial tests were predominantly obtained from animals exposed to the European endemic strain, BTV-8. Whether contesting or confirming the fitness for these assays, all the above-cited challenges must be carefully considered. For judicious selection of a group assay for a particular purpose and as part of the intra-laboratory verification process, recalibration of the working conditions fit for the population under study may be required.

False positive results caused by EHDV and other non-BTV antibodies occurred in all assays tested. These findings emphasize the need for caution when interpreting results and for additional diagnostic protocols to be used to differentially diagnose BTV in different populations. When compared with the cELISAs, the sELISA demonstrated the highest false positive rate for sera containing EHDV antibodies and lowest false positive rate with other non-BTV orbivirus antisera. The false positive rate may be explained by EHDV being the closest phylogenetic relative among orbiviruses to BTV [45], making it reasonable to suppose that VP7 from EHDV and BTV could share one or more epitopes similar enough for cross-reactive antibodies to be generated in infected hosts. A multiplicity of such epitopes would further raise the likelihood of false positive results using the sELISA.

In contrast, the cELISA is dependent on a single epitope which, if sufficiently different in EHDV from the epitope which elicited the competing mAb, would lessen or eliminate the occurrence of false positives. Notably, the Warrego virus sera were the most cross-reactive in the cELISAs, exhibiting elevated levels of inhibition. Benaganhalli et al. (2014) [25] report the Warrego virus VP7 sequences share the greatest homology with BTV and EHD VP7. It is therefore reasonable to presume that homology in the region bound by the competing mAb is high or identical across the VP7 epitope. The presence of competing antibodies in the cELISAs is also expected to be present in the sELISA. Still, their detection in the indirect (sELISA) system may be dose-dependent, since the test serum dilution for the cELISA is 1/10, compared to 1/100 for the sELISA. In other words, avidity makes EHD more cross-reactive in the sELISA, while affinity makes cELISAs formats more cross-reactive for other orbiviruses. Interferences caused by the presence of antibodies to related orbiviruses in shared ecologic and epidemic niches of these orbiviruses with BTV, EHD included, are at least suspected if not highlighted by the confirmed differences in how the BTV endemic negative samples and exotic negative samples reacted in both the sELISA and the ACDP cELISA. It is also possible that non-neutralizing BTV antibodies are present in these animals at low levels; these, however, were expected to be detected by the sensitive detection limits characteristics of the sELISA.

Differences in ASe were noted between assays, though between the bovine and ovine sELISAs, where the bovine sELISA appeared more sensitive than any other test. In contrast, the ovine sELISA was less sensitive than the IDEXX test in both sera and less sensitive than the ACDP test with one of them. These differences may relate to the immunodominant antibody responses of the two species relative to the specific epitope recognized by the competing monoclonal antibody used in the assays. Differences in the sequence or structure of that epitope between BTV strains and serotypes may also contribute.

When choosing to use a cELISA, it is crucial to ensure broad detection if considering fitness for the purpose of the testing lab and in the context of the country in question. The best approach is to use a front-line test, which is the most sensitive, and then re-test all negative samples on another test with differing characteristics. This practice of re-testing ensures that no potential positive cases are missed, thereby enhancing the overall detection capability. The use of SNT should always confirm positive detections and assist in surveying circulating serotype strains.

The preference for competition or blocking formats over direct and indirect ELISA formats may result from the increased ability of these platforms to better recognize and detect antibodies of classes other than IgG (e.g., IgM or IgA). Since the secondary conjugated antibodies of choice can recognize the species-specific immunoglobulin classes’ shared Fab regions, the sELISA should maintain its sensitivity characteristics independent of immunoglobulin class.

Overall, this study presented the development, evaluation and validation of a BTV sandwich ELISA, whose performance indicators approached those of other cELISAs and demonstrated a much-improved capability to detect antibodies to BTV-15 and to other serotypes, with no changes in sensitivity characteristics. It also evidenced the unique potential, but also drawbacks, of different ELISA formats. It showed that no one test is ‘fit for all’. Some can be more sensitive or specific than others in different contexts. Choice of a particular assay system must consider the inherent characteristics of the assays, the species of host animal and population under study, the presence of related off-target organisms, seasonal changes in the ecological systems under which diagnostic challenges occur, and, since no one size fits all, the best approach is perhaps using two of them.

## Figures and Tables

**Figure 1 viruses-16-01810-f001:**
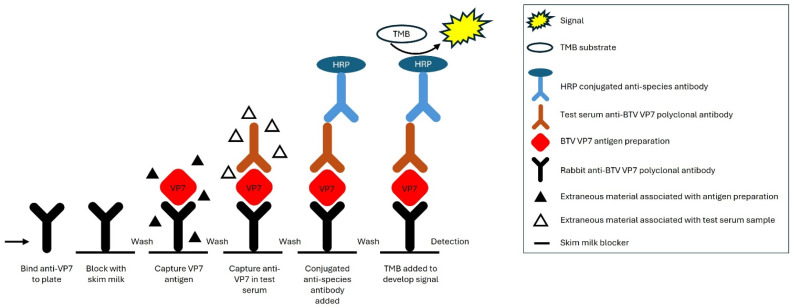
Principle of the sandwich ELISA. Rabbit anti-BTV VP7 polyclonal antibody is used to coat an ELISA plate. The plate is blocked with skim milk to prevent binding of potentially cross-reactive extraneous material (e.g., impurities in the antigen preparation or test serum proteins), so that only BTV VP7 antigen is bound via specific interaction with the coating antibody. The test serum is added allowing antibodies reactive to BTV VP7 in the sample to bind to the captured VP7 antigen. HRP-conjugated anti-species antibody (e.g., anti-bovine or anti-ovine antibody) is added and binds to the test sample-derived antigen-bound antibodies. Detection is achieved after addition of TMB substrate.

**Figure 2 viruses-16-01810-f002:**
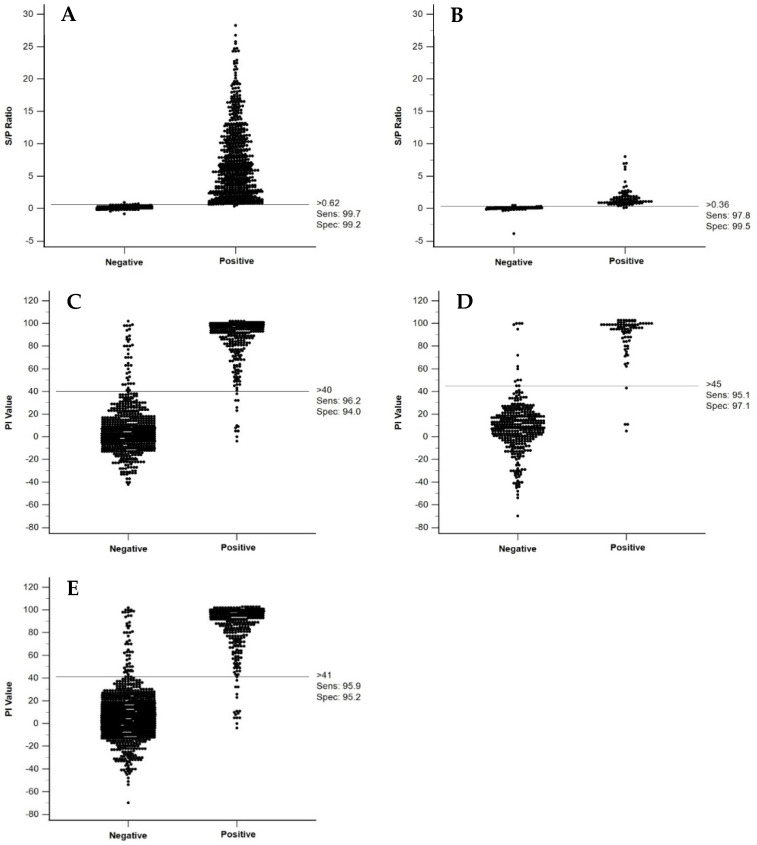
Interactive dot diagrams for anti-BTV antibody from bovine and/or ovine negative and positive populations as determined using the sELISA or ACDP cELISA. S/P ratio values obtained by testing bovine (**A**) or ovine (**B**) sera using the sELISA. PI values obtained by testing bovine (**C**), ovine (**D**), or bovine and ovine sera combined (**E**) from the ACDP cELISA. DSe (Sens) and DSp (Spec) values are indicated. Thresholds are indicated by the horizontal lines.

**Figure 3 viruses-16-01810-f003:**
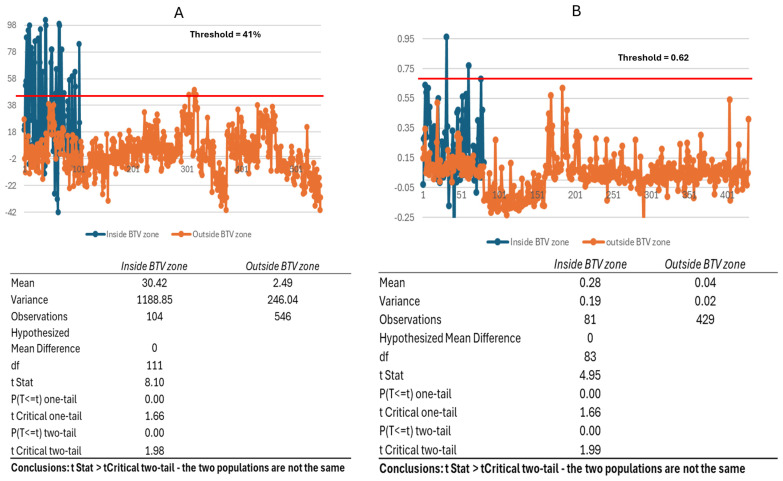
Negative bovine population sera from within (blue) or outside (orange) the Australian BTV exclusion zone were assessed using the ACDP cELISA (**A**) or the sELISA (**B**). Threshold values (red lines) for each assay are indicated and corresponding tabulated data for each test are presented. *t*-test: two-sample assuming unequal variances.

**Table 1 viruses-16-01810-t001:** Results of testing BTV-positive reference and experimental sera using 5 different ELISAs. The 3 panels of BTV-positive sera used in the study were of Australian (Australian serum panel) or South African (South African serum panel and CSL serum panel) origin. The serotype of each serum is indicated (column 1) with unique identifiers (column 2). The tests used to assess each of the sera were the sELISA, ACDP cELISA (ACDP), IDEXX cELISA (IDEXX), VMRD cELISA (VMRD) and IDVet cELISA (IDVet). Cut-off values for each test are indicated; Pos = positive, Neg = negative. Results of testing for the presence of anti-BTV antibodies are presented as positive (green), negative (red) or indeterminate (yellow). Corresponding test values are given as sample to positive control ratio (S/P) for the sELISA or percentage inhibition (PI) for the cELISAs.

		sELISA >0.62 ^1^ Pos >0.36 ^2^ Pos	ACDP >40 Pos	IDEXX ≤70 Pos >79 Neg	VMRD >60 Pos	IDVet <50 Pos
Serotype	Identifier					
Australian serum panel						
BTV-1	S13796 20/05/1983 bovine	28.28	100	4.48	93	4.51
BTV-2	2010 prototype isolate: 10-70368-15 (1080) ovine	0.99	99	12.26	85	9.26
BTV-3	20DPI 27/12/1992 ovine	0.74	97	14.97	60	11.45
BTV-5	BTV-5. VNT characterised field NT 15/362 3 bovine	1.46	83	5.24	94	4.63
BTV-7	2007 prototype isolate: ovine	0.78	96	8.01	94	5.8
BTV-9	S15195 08/02/1993 ovine	1.06	100	7.44	87	6.54
BTV-15	Isolate DPP0192 S13875 B71 27/04/1990 ovine	7.58	32	82.77	21	100.6
BTV-16	S15197 08/02/1993 ovine	2.90	98	10.51	82	5.23
BTV-20	S13793 V29 27/12/1992 bovine	21.92	100	4.27	90	4.29
BTV-21	S12207 E958 13/12/1982 bovine	11.30	97	3.84	93	4.31
BTV-23	B591 field NT 25/01/2010 ovine	1.86	98	4.01	92	4.46
South African serum panel						
BTV-1	35DPI 21/12/1993	1.11	99	4.82	92	5.14
BTV-2	35DPI 21/12/1993	1.05	99	10.64	89	5.66
BTV-3	35DPI 21/12/1993	1.19	95	13.2	84	8.33
BTV-4	35DPI 21/12/1993	1.06	100	4.24	93	4.49
BTV-5	35DPI 21/12/1993	5.80	99	11.33	91	5.78
BTV-6	35DPI 21/12/1993	1.19	99	8.24	92	4.67
BTV-7	35DPI 21/12/1993	0.79	95	5.44	91	0
BTV-8	35DPI 21/12/1993	1.18	99	9.29	93	5.92
BTV-9	35DPI 21/12/1993	0.82	97	4.49	90	7.19
BTV-10	35DPI 21/12/1993	1.23	99	11.32	93	4.96
BTV-11	35DPI 21/12/1993	0.97	99	9.76	93	5.62
BTV-12	35DPI 21/12/1993	1.08	99	12.1	92	6.03
BTV-13	35DPI 21/12/1993	1.68	71	54.81	65	6.29
BTV-14	35DPI 21/12/1993	1.10	95	22.61	89	7.88
BTV-15	35DPI 21/12/1993	0.94	43.6	70.6	22	55.2
BTV-16	35DPI 21/12/1993	1.15	99	13.22	91	5.02
BTV-18	35DPI 21/12/1993	4.01	101	9.99	93	6.11
BTV-19	35DPI 21/12/1993	0.98	98	5.18	93	54.83
BTV-22	35DPI 21/12/1993	0.71	93	4.34	85	12.74
BTV-24	35DPI 21/12/1993	0.85	97	22.86	94	7.64
CSL serum panel						
BTV-1	24/06/1985	2.81	102	4.5	92	4.79
BTV-2	24/06/1995	2.52	101	17.84	77	4.26
BTV-3	25/06/1985	0.73	101	10.06	65	4.31
BTV-4	25/06/1985	2.43	97	4.58	84	7.07
BTV-5	26/06/1985	1.25	101	14.49	11	43.54
BTV-6	26/06/1985	1.95	91	15	11	6.74
BTV-7	27/06/1985	1.89	79	76.7	80	18
BTV-8	27/06/1985	1.71	101	13.31	72	7.99
BTV-9	01/07/1985	1.36	101	39.47	84	5.39
BTV-10	01/07/1985	1.47	77	23.61	5	9.76
BTV-11	02/07/1985	1.12	101	9.19	75	28.9
BTV-12	02/07/1985	1.86	101	26.39	85	9.17
BTV-13	02/07/1985	2.29	94	41.66	88	14.92
BTV-14	03/07/1985	1.04	77	16.23	91	61.35

^1^ Cut-off S/P ratio for bovine sera. ^2^ Cut-off S/P ratio for ovine sera.

**Table 2 viruses-16-01810-t002:** Summary statistics for BTV antibody positive or negative ovine or bovine sera assessed using the sELISA or ACDP cELISA. Results are expressed as sample to positive control ratio (S/P) for the sELISA and percentage inhibition (PI) for the ACDP cELISA. Data represented are number of positive or negative sera from each species tested, mean and median S/P or PI values obtained, standard deviation, minimum and maximum values, and interquartile range.

	sELISA				ACDP cELISA			
	Ovine Sera		Bovine Sera		Ovine Sera		Bovine Sera	
	Positive	Negative	Positive	Negative	Positive	Negative	Positive	Negative
No. of sera	91	440	661	509	81	385	314	650
Mean	1.69	0.02	7.36	0.07	89.71	7.56	83.13	6.96
Median	1.32	0.02	6.10	0.05	96.48	8.26	95	3.63
Standard deviation	1.46	0.20	5.53	0.16	19.48	23.89	21.24	22.37
Minimum	0.13	−3.89	0.35	−0.82	5.27	−189	−47	−42
Maximum	8.05	0.53	28.28	0.96	103.34	100	102	102
Interquartile range	0.98	0.06	7.88	0.11	11.27	19.65	16.5	9.59

**Table 3 viruses-16-01810-t003:** Relative analytical sensitivity of sELISA, ACDP cELISA (ACDP) and 3 commercial cELISAs (IDEXX, VMRD and IDVet). Limits of detection were determined using one BTV-3 and one BTV-7 Australian reference ovine and 3 Australian post-infection bovine sera, positive to BTV-16, BTV-1 and/or BTV-21, diluted to threshold levels of reactivity. Titers represent the dilutions of sera in addition to the routine starting dilution recommended for the assay (sELISA, 1/100; ACDP cELISA, 1/10; IDEXX cELISA, 1/5; VMRD cELISA and IDVet cELISA, 1/2). Titers are expressed as log_4_ values.

	sELISA	ACDP	IDEXX	VMRD	IDVet
Ovine serum 1	6.80	5.86	9.57	3.07	4.45
Ovine serum 2	22.20	25.29	31.08	16.80	10.00
Bovine serum 1	11.29	1.86	1.71	<1	<1
Bovine serum 2	24.63	4.27	13.35	4.11	6.75
Bovine serum 3	77.62	4.37	3.77	1.24	2.67

**Table 4 viruses-16-01810-t004:** Diagnostic sensitivity (DSe) and diagnostic specificity (DSp) estimates of the sELISA, ACDP cELISA (ACDP) and commercial cELISAs assays (IDEXX, VMRD and IDVet). Results for the sELISA and ACDP cELISA were obtained in-house. Results for the commercial cELISAs were taken from Niedbalski, 2011. DSe and DSp estimates were calculated for each assay using ovine or bovine samples separately. Estimates for the ACDP cELISA were also calculated using bovine and ovine sera collectively.

sELISA Ovine		sELISA Bovine		ACDP Ovine		ACDP Bovine		ACDP Combined		IDEXX		VMRD		IDVet	
DSe	DSp	DSe	DSp	DSe	DSp	DSe	DSp	DSe	DSp	DSe	DSp	DSe	DSp	DSe	DSp
97.8	99.5	99.7	99.2	95.15	97.1	96.2	94.0	95.9	95.2	82.80	100.00	69.50	99.30	96.50	99.30
In-house assessments										Niedbalski, 2011 [36]					

**Table 5 viruses-16-01810-t005:** Results of assay sensitivity testing for BTV-15 positive bovine and ovine reference sera. Sera were either of bovine or ovine origin as indicated. Positive (green), indeterminate (yellow) and negative (red) identifications and their corresponding test values are given as sample to positive control ratio (S/P) for the sELISA or percentage inhibition (PI) for the cELISAs. Cut-off values for each test are indicated; Pos = positive, Neg = negative. Total positive identifications for each assay are given as a percentage of all samples tested (bottom row). The tests used to assess each of the sera were the sELISA, ACDP cELISA (ACDP), IDEXX cELISA (IDEXX), VMRD cELISA (VMRD), IDVet cELISA (IDVet) and virus neutralization test (VNT). All sera were deemed to be positive for BTV-15 antibodies by VNT (column 7). N/A = not assessed.

Serum	sELISA >0.62 ^1^ Pos >0.36 ^2^ Pos	ACDP >40 ^1^ Pos ≥45 ^2^ Pos	IDEXX ≤70 Pos >79 Neg	VMRD >60 Pos	IDVet <50 Pos	VNT
2013 bovine	1.62	56	107.90	6	123.93	80
2013 bovine	1.76	−7	103.56	19	158.53	160
2013 bovine	2.41	36	137.09	25	140.74	320
2013 bovine	1.04	4	122.86	47	137.51	20
2013 bovine	3.55	94	113.27	−19	12.07	40
2013 bovine	1.13	−5	107.58	45	160.32	5
2013 bovine	3.57	88	22.44	19	53.69	160
2013 bovine	2.11	9	116.50	79	159.95	80
2013 bovine	2.17	87	74.97	44	90.72	80
2013 bovine	3.02	96	134.13	83	14.45	40
2013 bovine	2.50	98	31.69	−9	52.21	20
2013 bovine	1.12	65	97.10	−4	158.86	40
2019 bovine	7.60	14	96.48	N/A	37.7	20
Anti-BTV-15 Aus ovine	7.58	32	82.77	21	100.63	160
Anti-BTV-15 SA ovine	3.75	43	70.60	22	55.2	40
Positives (%)	100	53	13	14	20	100

^1^ Cut-off for bovine sera. ^2^ Cut-off for ovine sera.

**Table 6 viruses-16-01810-t006:** Results of assay sensitivity testing for BTV PCR positive bovine sera. Positive (green), indeterminate (yellow) and negative (red) identifications and their corresponding test values are given as sample to positive control ratio (S/P) for the sELISA, percentage inhibition (PI) for the cELISAs, or percent sample to negative control (% S/N) for serum neutralization test (SNT). Cut-off values for positive results for each test are indicated. Total positive identifications for each assay are given as a percentage of all samples tested (bottom row). The tests used to assess each of the sera were the sELISA, ACDP cELISA (ACDP), IDEXX cELISA (IDEXX), VMRD cELISA (VMRD), IDVet cELISA (IDVet) and SNT. All sera were deemed to be positive for BTV by PCR. N/A = not assessed.

Serum Identifier	sELISA S/P > 0.62	ACDP PI ≥ 40	IDEXX PI ≤ 70	VMRD PI ≥ 60	IDVet PI < 40	VNT
S1	0.91	86	109	26	5	Negative
S2	7.44	14	97	N/A	38	Positive
S3	12	87	135	72	18	Positive
S4	6.64	100	125	90	0	Indeterminate
S5	1.69	69	62	88	3	Positive
S6	0.91	−4	18	71	7	Positive
S7	6.64	100	125	90	0	Positive
S8	2.40	90	17	32	22	Positive
S9	1.17	82	80	59	21	Positive
S10	2.89	100	31	36	32	Positive
S11	5.44	49	15	55	6	Positive
S12	10.55	75	11	28	87	Positive
S13	5.54	38	48	36	31	Positive
S14	8.13	98	52	37	5	Positive
Positives (%)	100	79	57	38	93	86

## Data Availability

The original contributions presented in this study are included in the article/Appendix A. Further inquiries can be directed to the corresponding author(s).

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
