# Peer review of "Challenges of BTV-Group Specific Serology Testing: No One Test Fits All"

_viruses, 2024, doi:10.3390/v16121810_

Round 1
Reviewer 1 Report
Comments and Suggestions for Authors
Summary:
The authors assess the performance of a newly developed sandwich ELISA (sELISA) designed to detect antibodies to Bluetongue Virus (BTV) in bovine and ovine sera. The validation process compared this assay with four commercial competition ELISAs (cELISAs), using samples from both BTV-endemic and non-endemic regions. Results showed that the sELISA exhibited high diagnostic sensitivity (DSe) and specificity (DSp) across multiple BTV serotypes, including BTV-15, which is endemic to Australia. Notably, the sELISA outperformed the cELISAs in detecting antibodies against BTV-15 and demonstrated lower cross-reactivity with other orbiviruses, although it was unable to differentiate antibodies to epizootic hemorrhagic disease virus (EHDV). The manuscript underscores the importance of serological tests that can accurately detect a range of serotypes while minimizing false positives due to cross-reactivity with non-BTV orbiviruses. Overall, the study presents valuable findings for improving BTV serology testing, but enhancing the discussion of clinical and practical applications, as well as refining certain sections for clarity, would increase its impact.
Suggestions
-
While the manuscript focuses on the technical performance of the sELISA, a more in-depth discussion of its clinical and epidemiological significance would strengthen the paper.
-
Including a dedicated section outlining the limitations of the sELISA, particularly in detecting certain BTV strains and addressing cross-reactivity with non-BTV orbiviruses, would provide a more comprehensive analysis.
-
Some of the statistical comparisons, such as the ROC curve analysis and sensitivity/specificity metrics, would benefit from clearer visual aids like bar graphs or heat maps to better illustrate the performance differences between the tests.
-
The figure and table legends are somewhat brief and would be improved by providing more detailed descriptions.
Reviewer 2 Report
Comments and Suggestions for Authors
The manuscript by Di Rubbo et al describes the development of a new sandwich ELISA that was compared and verified with other existing ELISA protocols using a big set of samples.
The study is extensive and well-designed for thorough evaluation of the new ELISA and a comparative evaluation of the performance of different assays.
Authors have made fantastic effort in elaborating the pros and cons of each assay in detecting the different serotypes of BTV and specifically BTV15.
minor comments:
1) there are few incomplete sentences in the manuscript: line 96, line 106
2) introduction: please include a brief introduction to immune sero-dominant epitopes/viral proteins that are commonly targeted for ELISA development
Reviewer 3 Report
Comments and Suggestions for Authors
Generally, this manuscript is well-written but can be improved through minor revisions. Most of the tables and figures are data- and calculation-dense. More detailed captions and footnotes would make them more clear.
Line 98 – A more detailed description of the process should be included. The text description is good, but the Figure should stand alone.
Line 106 – Is there missing text?
Line 124 -129 – Perhaps re-align the Table.
Line 226 – The Table description is very incomplete, again, it should stand alone.
Line 287 – Avoid the use of the terms “show” or “exhibit”. Just state the results.
Comments on the Quality of English LanguageOnly minor editing and revision is required.
